# Molluscivorous red knots rapidly adjust to a plant diet

Marie De Wilde[1],*, Berber Maarsingh[2], Luc G. G. de Monte[3], Anne Dekinga[3], Allert I. Bijleveld[3], Hidde Kressin[3], Aldert L. Zomer[4] and Theunis Piersma[2,3,5]

## ABSTRACT

Dietary flexibility is key to adjusting to environmental change. In Mauritania, the seemingly obligatory molluscivorous red knots *Calidris canutus* were observed to eat seagrass rhizomes. To study the ability of knots to live on plant material, we performed a diet-change experiment on captive individuals. Two groups of five were fed protein-rich pellets for 13 weeks, then plant-based pellets for 6 weeks, then reversed back to protein-rich pellets for 4.5 weeks. During the first days following the shift to the plant diet, body mass declined by 14% before increasing and stabilizing to lower levels. Faecal colour changed from green (i.e. gall, suggesting starvation) to brown and was produced in larger quantities when the birds ate plant pellets. These experimental data prove that knots can indeed live on a plant-based diet, with the observed changes suggesting that adjustments of the digestive system, i.e. gut morphology and microbiome, take only a few days.

KEY WORDS: Red knot, *Calidris canutus*, Dietary flexibility, Diet-change experiment, Adaptation

## INTRODUCTION

Dietary flexibility is key to adapting to rapid environmental change, enabling individuals to cope with short-term variations in food availability (Dunham, 2017; Spencer et al., 2017). Many species have physiological and morphological adaptations that enable them to efficiently capture, process and digest specific prey (Stevens and Hume, 1995b). However, these adaptations may constrain their ability to shift to alternative resources that require fundamentally different digestive strategies (Futuyama and Moreno, 1988; Hecker et al., 2019).

Migratory members of the shorebirds (Charadrii) are a well-studied group that encounter substantial seasonal and geographic variation in prey type and availability (Angarita-Báez and Carlos, 2024; Battley and Piersma, 2005; Hall et al., 2021; van de Kam et al., 2004). Many species adjust their diet accordingly and feed on insects on their breeding grounds and molluscs or crustaceans on coastal mudflats on their non-breeding grounds (Angarita-Báez and

[1]Department of Marine Microbiology and Biogeochemistry, NIOZ Royal Netherlands Institute for Sea Research, Den Burg, 1790 AB Texel, The Netherlands. [2]Faculty of Science and Engineering, University of Groningen, 9700 AB Groningen, The Netherlands. [3]Department of Coastal Systems, NIOZ Royal Netherlands Institute for Sea Research, Den Burg, 1790 AB Texel, The Netherlands. [4]Department of Infectious Diseases and Immunology at the Faculty of Veterinary Medicine, Utrecht University, 3508 TC Utrecht, The Netherlands. [5]BirdEyes, Centre for Global Ecological Change at the Faculties of Science and Engineering and Campus Fryslân, University of Groningen, 8911 CE Leeuwarden, The Netherlands.

*Author for correspondence (marie.de.wilde@nioz.nl)

M.D.W., 0009-0003-1938-6177

Carlos, 2024; Skagen and Oman, 1996). These shifts mostly remain within the invertebrate prey category which is relatively highly digestible and rich in energy (Karasov and Douglas, 2013). The inclusion of non-animal material in shorebird diets is usually limited to seasonal supplements, likely to reduce competition for resources (Hall et al., 2021; Huang et al., 2022; Lourenço et al., 2010; Robin et al., 2013). Plant-based diets are typically low in protein and high in fibre (Klasing, 1998a), requiring extended ingesta retention time and microbial fermentation (Stevens and Hume, 1995a). Shorebird species with a uniquely plant diet are absent, likely because the necessary traits are associated with increased gut volume and body mass, potentially compromising flight efficiency (Klasing, 1998b; Mathot et al., 2020; McWhorter et al., 2009; Pennycuick, 1968; van den Hout et al., 2010). Many shorebirds exhibit substantial physiomorphic flexibility and can regulate fat stores for migration and adjust the size of digestive organs in response to diet type, quantity and intake rate (Battley and Piersma, 2005; Piersma et al., 2021). This organ plasticity allows for short-term flexibility and energy optimization but may not extend to the efficient digestion of fibrous plant material.

Red knots (*Calidris canutus*) are one of the dietary more specialized migratory shorebirds (Piersma, 2007). They have a unique sensory capacity to detect hard-shelled molluscs in soft sediments (Piersma et al., 1998) and a specialized gut to crush ingested hard-shelled prey in muscular gizzards and cope with the abrasive shell remains in the intestine (Battley and Piersma, 2005; van Gils et al., 2005; Yang et al., 2013). Surprisingly, on the seagrass beds of the Parc National du Banc d'Arguin in Mauritania, West Africa, red knots frequently consume seagrass rhizomes (*Zostera noltii*), especially in years when the preferred bivalves are thinly spread or buried out of reach for their 3.5-4 cm long bill (Oudman et al., 2018; van Gils et al., 2016). In the Wadden Sea, juvenile *islandica* knots have been observed eating green algae (Ersoy et al., 2024).

The conundrum that a molluscivorous shorebird includes individuals that mostly eat plant material raises a fundamental question: how do such highly specialized animals cope with a diet so radically different from their norm? In this study, we explored the ability of red knots to feed on plant material by confronting captive individuals with a diet switch from protein-rich fish pellets to pellets that are fully plant based, and in reverse. To investigate whether red knots can sustain themselves on a plant-based diet, we monitored (1) the body mass of red knots on a plant-based diet and compared this to body mass predictions based on historical data of individuals on a protein-rich diet under similar conditions, and (2) collected faecal output as a non-invasive metric for digestive function. We predicted that, after an adjustment period of a couple of days (Dekinga et al., 2001), body mass would be similar to what is expected for that time of the year, and that faecal output would show no abnormalities.

## RESULTS

### Body mass

Throughout the study, body mass remained within the historically measured biological variation of red knots living in the shorebird

Biology Open

facilities (Fig. 1A). Both experimental groups show nearly identical patterns in body mass change despite one group being on average 4.73 g (s.e.m.=0.69) heavier than the other. In the first 3 days after the switch to a plant-based diet, all birds lost on average 14% (s.e.m.=0.78) of their body mass, but no individual reached the critical limit of a 100 g. In the weeks thereafter, body mass values increased before stabilizing at a level that was lower than before the experimental perturbation. Interestingly, when shifting back to the control diet, body mass spiked, with birds increasing on average 13% (s.e.m.=1.81) in body mass in only 3 days.

Fig. 1B shows the observed body mass compared to the predicted body mass modelled from their body mass trajectory and historical data, from the start of the vegetarian period onwards. Red dots show the days on which body mass deviates with more than 1 s.d. from what is expected for the same individuals had there been no plant-based diet switch. At the start of the switch to the plant-based diet,

body mass appears to be notably lower for 20 days. During the second half of the experimental period, body mass returns to equilibrium and deviates less than 1 s.d. from the prediction and thus seems to be within the expected range during that time of the year. The opposite is observed when switching back to the control diet, with deviations of more than 1 s.d. higher than expected. Body mass stabilizes to normal levels after 1 week (for Z-scores, see Table S1).

## Faecal matter

Faeces were green in colour and contained less material at the start of the plant diet. During the first week, they shifted from looking like an undigested pellet to a more consistent but wet dropping, similar to droppings observed in the wild. Although we were not able to quantify the droppings, we observed that birds produced noticeably larger quantities of faecal matter during the plant-based period compared to the control diet. Fig. 2A shows the daily change in body

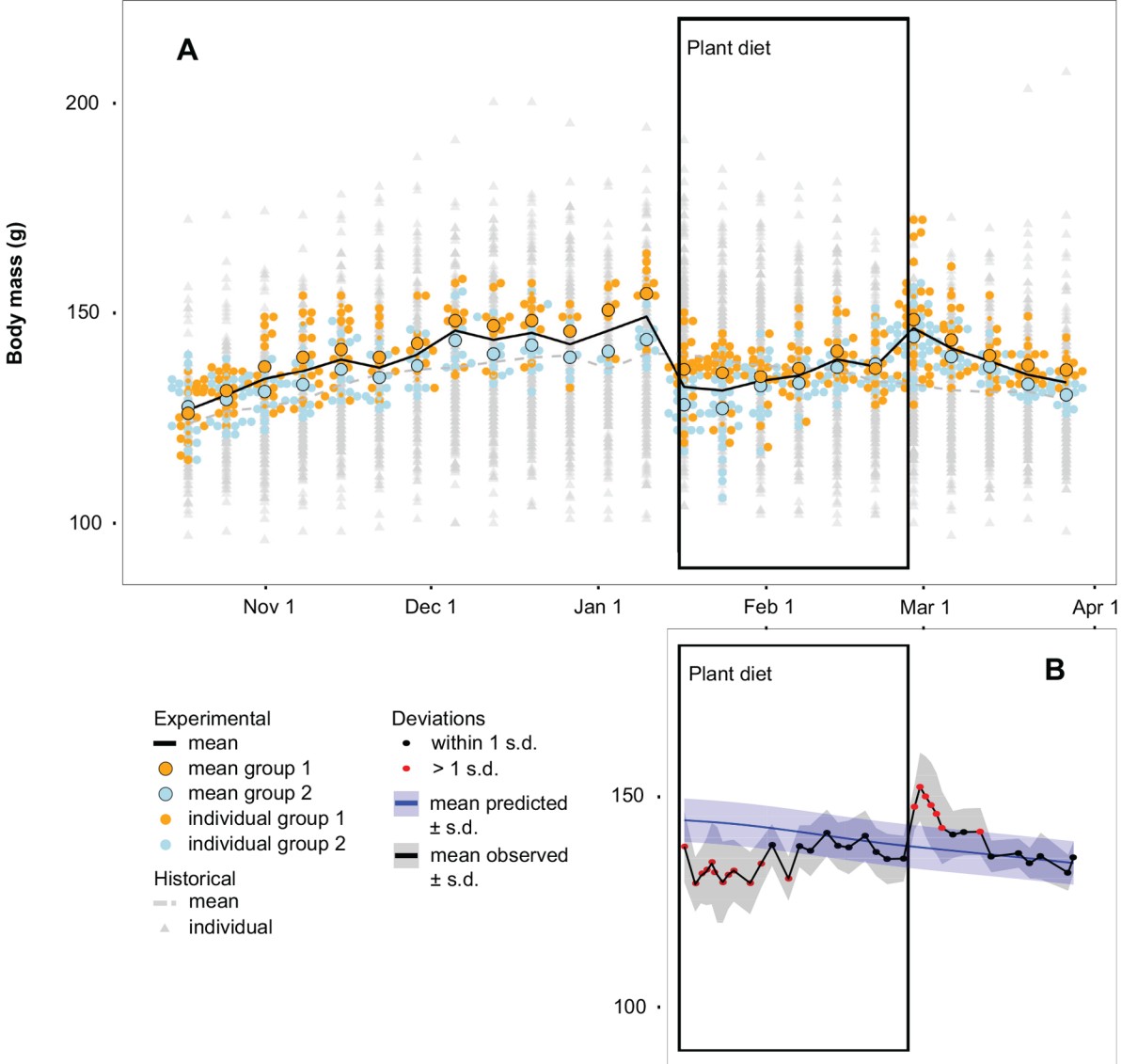

**Fig. 1. Body mass during the experimental period of red knots in captivity.** (A) Weekly mean body mass (g) during the experiment (solid black line) in 2023 (*n*=10). Larger coloured dots represent the mean weekly body mass of group 1 (orange) and group 2 (blue). Individual body mass measurements are represented in smaller dots. The mean (grey dashed line) and individual datapoints (grey triangles) of historical data (from 2017 to 2021) are included for comparison (*n*=257). (B) Mean body mass (black) on each sampling day. Measurements from the vegetarian period onwards are compared to what is expected (blue) of red knots had they been fed the control diet. Deviations higher than 1 s.d. are shown in red. Black points indicate deviations within 1 s.d.

mass for each individual and the colour of their respective droppings collected that day. From day 4 onwards, body mass increased and stabilized around a new, lower value, with some individual variation in the time until stabilization. Faecal colour closely followed the observed changes in body mass. Fig. 2B shows the decrease in relative amount of green colour detected in the faecal samples as body mass stabilized. These results are in line with the live observations and video footage, showing that the birds were hesitant to eat the plant-based pellets at first. As soon as one individual initiated feeding, the rest of the group followed soon after (around day 3).

## DISCUSSION

This study shows that red knots in captivity can survive on a plant-based diet. Video footage in combination with the green colour of the faeces in the first 3 days after the diet switch indicate that the initial drop in body mass is likely due to starvation only. The changes in faecal properties observed after resumption of feeding indicate an adaptive physiological change in the gastrointestinal tract to the less nutritious, high fibre diet. The increased faecal output suggests that they ate higher volumes to make up for the lack of nutritional quality (Stevens and Hume, 1998). The body mass trajectory (Fig. 1) indicates a total recovery period of 2 weeks. The flexibility in digestive organ size could explain why red knots are

able to sustain themselves on a plant-based diet as an increased gut size would prolong food retention time, a typical adaptation in strict herbivores (Stevens and Hume, 1998). However, it seems unlikely that the initial, rapid adjustment to the high fibre diet is solely the result of a change in organ size as this usually takes longer than 3 days (Dekinga et al., 2001; Piersma et al., 1996). Research shows that house sparrow nestlings can modulate intestinal enzyme activity within 24 h following an abrupt dietary change (Brzęk et al., 2009; Rott et al., 2017). Although studies in shorebirds are absent, similar adjustments in digestive enzyme production could help explain the timespan in which red knots adjust to a sudden switch to a plant-based diet, however not the reverse, as we would expect to also observe an adjustment period with the diet switch from plant to fish pellets (Brun et al., 2021). Instead, body mass was disproportionally high for the season and stage, and there was no notable change in faecal output when shifting back to the control diet. Lastly, animals host microbial communities in their gastrointestinal tract (i.e. the gut microbiome) aiding digestion. A compositional change in gut microbiome can occur in a matter of days in response to diet and environmental change (Bodawatta et al., 2022; Grond et al., 2018; Hamaker and Tuncil, 2014; Kuziel and Rakoff-Nahoum, 2022; Zhang et al., 2021). This is now a work in progress.

## MATERIALS AND METHODS
### Animals and housing
Ten red knots (*Calidris canutus islandica*) were captured on 16 October 2023 at the islet of Griend (53°15′N, 5°15′E) in the Dutch Wadden Sea, using mist nets. Upon capture, each individual was measured, ringed with a unique colour combination for identification and had a blood sample collected for sex determination, as part of an ongoing program. The birds were initially held in small enclosures and transported to the experimental facilities at the Royal Netherlands Institute for Sea Research (NIOZ), Texel (53°00′13″N, 4°47′23″E) on 18 October 2023. All procedures were conducted under the project license granted to NIOZ by the Centrale Commissie Dierproeven (license number AVD80200202215943). Experimental protocols and animal handling were approved by the institutional animal welfare body (approval number NIOZ-2023-03) and conducted in accordance with the Dutch legislation on animal experimentation (Wet op de Dierproeven).

Birds were housed in custom-built aviaries designed to replicate natural environmental conditions while maintaining experimental control (Milot et al., 2014). Each aviary (4.5 m length×1.5 m width×2.3 m height) included wire-mesh windows that allowed exposure to natural airflow, ambient temperature, and photoperiod, while protecting the birds from rain and predators. Adjacent aviaries were separated by solid partitions, ensuring both visual and physical isolation, though auditory contact remained possible. A separate water system delivered flowing saltwater (sourced from the nearby Wadden Sea) across soft sediments and the aviary floor, reducing the risk of foot infections. Freshwater was provided via an overflowing basin, and food was offered *ad libitum* in two identical trays per aviary. The standard captive diet (hereafter, control diet) consisted of commercial pelleted food rich in animal protein (see Table 1).

### Experimental setup and sampling
The experiment was conducted between 18 October 2023 and 28 March 2024. Upon arrival, the ten red knots were randomly assigned to two identical groups for housing purposes, using Microsoft Excel (Version 2309). Sample size was not calculated *a priori*, as this study was intended as a pilot. The number of animals was determined based on prior experience with this species at the facility. Each group consisted of five individuals and received the same treatment throughout the study. To facilitate acclimatization, each group was accompanied by two experienced red knots that were already adapted to the aviary environment and pelleted diet. These 'trainer' birds remained with their respective groups only during the initial acclimation phase of 11 weeks and were excluded from

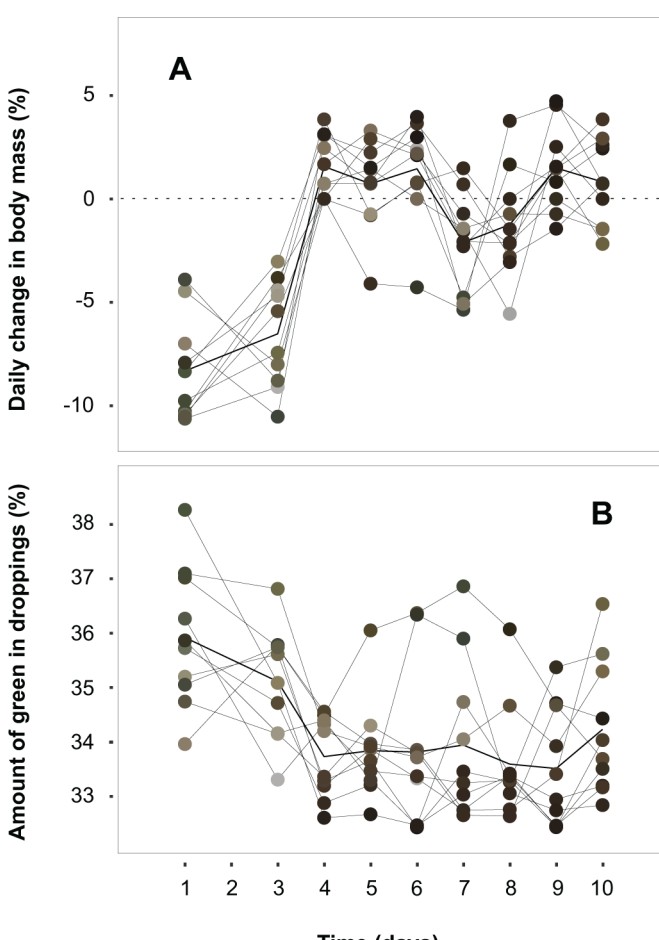

**Fig. 2. Body mass change and faecal colour during the first 10 days following the plant-based diet switch.** The coloured dots show the individual measurements in the respective faecal colour. The solid black lines are the mean. (A) Individual measurements of body mass change (%). (B) Percentage of green colour measured in the faecal samples.

**Table 1. Components of Trouvit, a commercial fish feed formulated for carnivorous fish species and complete rabbit pellets, typically used for small herbivores**

|  | Control diet: Trouvit (Skretting, Norway) | Experimental diet: rabbit pellets (BENELUX, Belgium) |
|---|---|---|
| Ingredients | Soybean meal, fishmeal, wheat, haemoglobin powder (spray dried), hydrolysed feather meal, wheat gluten, soluble byproducts of corn starch production, fish oil, poultry oil, soy protein concentrate | Fine wheat bran, lucerne meal, sunflower seed meal, dried (sugar) beet pulp, barley bran, (sugar) cane molasses, palm kernel meal, hazelnut expeller, wheat, maize, calcium carbonate, monocalcium phosphate, sodium chlorate |
| Component | Relative abundance (%) | Relative abundance (%) |
| Crude protein | 47.0 | 16.0 |
| Crude fat | 16.0 | 2.9 |
| Crude fibre | 2.5 | 15.5 |
| Crude ash | 7.0 | 8.5 |

experimental treatments thereafter for ethical reasons. They were removed 2 weeks before the start of the experimental treatment to eliminate a potential carry-over effect. The experiment was structured into three distinct phases. The first and third phases had a duration of 13 and 4.5 weeks, respectively, and served as baseline and control periods, during which all experimental birds were fed the control diet. In the second, experimental phase, birds received a plant-based diet for 6 weeks composed exclusively of plant material (see Table 1). Food was provided *ad libitum* throughout all phases, and group assignments remained consistent for the duration of the study.

To ensure animal welfare throughout the experiment, daily visual health assessments and weekly full health checks were conducted by trained animal caretakers (see Milot et al., 2014). From the onset of the experimental (plant-based) diet phase, continuous live video monitoring was implemented. This allowed remote observation of activity, social behaviour, and feeding. Red knots can lose up to 15 g/day when not feeding (Dietz and Piersma, 2007). Body mass was measured by putting the individual in a plastic tube onto a portable scale (OHAUS CS2000, Parsippany, NJ, USA). This was done on average three times per week, except during the first week following each dietary transition, when sampling was performed every day. Below a critical body mass threshold of 100 g an individual was considered at risk of starvation (Dietz and Piersma, 2007) and would have been removed from the experimental group. This did not happen, and the behaviour of the birds in the aviaries was normal for red knots in captivity.

Additionally, faecal samples were collected at the same frequency as body mass measures and used as a non-invasive metric for digestive function. Faecal consistency, colour, and frequency provide insight into gastrointestinal health (Atkinson et al., 2020; Rose et al., 2015). Healthy faeces in red knots typically consist of one cohesive dropping, cylindrical in shape and coloured in various shades of brown. Green colouration in bird faeces can be an indicator of fasting, due to a high concentration of biliverdin, a bile pigment normally diluted in regular excreta (Mateo et al., 2004). To collect faecal samples, birds were placed in a custom-built box divided into individual compartments. In such conditions, birds typically defecated every 10–30 min (Onrust et al., 2013). Droppings were observed before being collected and stored for further analysis.

We were unable to measure the amounts of food that the red knots were eating. Apart from the difficulty of assessing intake rate of individual intakes in a group, red knots often spill food before eating it. Due to the necessary wet floors of the aviaries (to keep the feet of the birds clean and intact) spilled food quickly dissolves and gets washed away with the flowing salt water. Photographs of the faecal samples were taken using a portable lightbox (ZoeZo Design 40×40 cm, LED lighting, The Netherlands) with a Nikon D3400, using a 40 mm lens at f/7.1, 1/60 s ISO, 100 with no flash. Digital images were analysed using ImageJ software (version 1.54 g), where the outline of each dropping was manually selected. The mean red (R), green (G), and blue (B) pixel values were extracted for each sample, and corresponding colour codes were calculated based on the RGB values and converted to HEX codes for quantitative colour analysis.

The lack of a parallel control forms a limitation in this study. Relying on the historical data has implications in that individual body mass trajectories are difficult to capture in a model.

## Statistics

A predictive generalized additive mixed effects model (GAMM) was developed using historical data from 2017-2021 of *islandica* red knots previously housed under similar environmental and seasonal conditions. Data were filtered to remove individuals that were not held in captivity throughout the three phases. The model was internally evaluated using the leave-one-year-out-validation and applied to the current dataset to predict the body mass of the individuals from the start of the vegetarian period onwards (Fig. S1). Residuals, calculated as the difference between observed and predicted values, were analysed using a linear regression model to indicate deviations from the expected body mass. Standardized residuals (Z-scores, see Table S1) were then calculated to quantify how mean observations deviated from model expectations in a biologically meaningful way.

## Software

Plots and analyses were performed using R (version 4.4.1) using the packages dplyr, mgcv, ggplot2, lme4, tidygam, ggbeeswarm, lubridate, lmerTest, emmeans, broom.mixed, car, Metrics, nlme and purrr. ChatGPT (GPT-5.2, OpenAI, 2025) was used to assist with R code and grammar and spelling of the text. Adobe Illustrator was used for figure layout and labelling.

## Acknowledgements

We thank Tim Oortwijn and Job ten Horn for assistance with sample collection and bird maintenance; and Laura Villanueva, Jaap Wagenaar for reviewing and editing the manuscript. In addition, we thank Edwin Keijzer for designing and making the sampling crates; and Julia Engelmann and Harry Witte for help with understanding GAMMs. We also thank everyone who has helped with the maintenance and data collection in previous years, in particular, Job ten Horn, Eva Kok, Kasper van Kraaij, Maureen Sikkema, Thomas Lameris, Evelien Witte, Kim Mathot, Selin Ersoy and Martin Bulla.

## Competing interests
The authors declare no competing or financial interests.

## Author contributions
Conceptualization: M.D.W., A.D., A.I.B.; Data curation: M.D.W., B.M., L.G.G.d.M., A.D., H.K.; Formal analysis: M.D.W., A.L.Z.; Methodology: M.D.W.; Supervision: T.P.; Visualization: M.D.W.; Writing – original draft: M.D.W., T.P.; Writing – review & editing: B.M., L.G.G.d.M., A.D., H.K., A.I.B., T.P., A.L.Z.

## Funding
This research was funded by Universiteit Utrecht [UU-NIOZ no. NZ4543.24]. Partial funding was provided from the operating budget of the Koninklijk Nederlands Instituut voor Onderzoek der Zee (NIOZ) Department of Coastal Systems. Open Access funding provided by Universiteit Utrecht. Deposited in PMC for immediate release.

## Data and resource availability
All relevant details of resources can be found within the article and its supplementary information. The data supporting the findings of this study are available at Dryad (doi:10.5061/dryad.wdbrv1645).

## Peer review history
The peer review history is available online at https://journals.biologists.com/bio/lookup/doi/10.1242/bio.062365.reviewer-comments.pdf

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

Biology Open