## [Peer Review File · Biology Open]

Molluscivorous red knots rapidly adjust to a plant diet

Marie De Wilde, Berber Maarsingh, Luc de Monte, Anne Dekinga, Hidde Kressin, Allert I. Bijleveld, Theunis Piersma and Aldert L. Zomer

DOI: 10.1242/bio.062365

Editor: Lewis Halsey

Review timeline

Original submission:	13 October 2025
Editorial decision:	22 October 2025
Resubmission received:	12 November 2025
Editorial decision:	20 November 2025
First revision received:	18 February 2026
Accepted:	2 March 2026

Original submission

First decision letter

MS ID#: bio.062314

MS Title: Molluscivorous red knots rapidly adjust to a plant diet

Authors: Marie De Wilde; Berber Maarsingh; Luc de Monte; Anne Dekinga; Hidde Kressin; Allert I. Bijleveld; Theunis Piersma; Aldert L Zomer

I am writing to let you know that I have now reached a decision on the above manuscript. I am afraid that, after careful consideration, I feel that it cannot currently be accepted for publication in Biology Open.

A main claim of the manuscript is that body mass declines considerably but then increases before then stabilising. But based on Figure 1 this isn't what happens, rather body mass decreases, then increases but then decreases again and ultimately the birds are somewhat lighter than before the experimental perturbation. Figure 2 seems to indicate that body mass is lost and then it stabilises at a new lower value. The birds may be able to survive, therefore, on the plant-based diet but it is unclear that they thrive. A subtler consideration of the body mass findings coupled with additional data types e.g. representing the microbiome, could make your manuscript very suitable for submission to BiO.

I realise that this is disappointing news, but I do hope you find my comments helpful.

Resubmission

First decision letter

MS ID#: bio.062365

MS Title: Molluscivorous red knots rapidly adjust to a plant diet

Authors: Marie De Wilde; Berber Maarsingh; Luc de Monte; Anne Dekinga; Hidde Kressin; Allert I. Bijleveld; Theunis Piersma; Aldert L Zomer

I have now reached a decision on the above manuscript.

The reviewer reports are shown at the bottom of this email.

As you will see, the reviewers raised a number of substantial criticisms that prevent me from accepting the paper at this stage.

They suggest, however, that a revised version might prove acceptable, if you can address their concerns. If you think that you can deal satisfactorily with the criticisms on revision, I would be pleased to see a revised manuscript. We would then return it to the reviewers.

At this stage, we also ask you to ensure your manuscript complies with our formatting guidelines. Provided you are able to fully address the referees' comments, we are positive about publication of your paper (we accept over 95% of revision submissions) and therefore hope you won't mind any extra work involved in reformatting your manuscript at this point.

Please upload both a 'clean' version of your Word file, along with a highlighted version clearly showing where you have made changes in the revised manuscript. Please avoid using 'Track changes' in Word files as these are lost in PDF conversion.

I should be grateful if you would also provide a point-by-point response detailing how you have dealt with the points raised by the reviewers in the 'Response to Reviewers' box. Please attend to all of the reviewers' comments. If you do not agree with any of their criticisms or suggestions please explain clearly why this is so.

Reviewer 1

Comments for the author

The authors exposed a group of typically molluscivorous red knots (*Calidris canutus islandica*) to a primarily plant based diet. They reported that the studied birds rapidly adapted to the new diet, demonstrating dietary flexibility in this species.

Overall I found this to be a very interesting study and believe it is an important topic to explore, considering the important role that dietary flexibility may play in this rapidly changing world. The manuscript is generally well-written and parts are compelling. There were, however, some aspects that I found difficult to understand and in my opinion, the manuscript would be strengthened by clarification of some core concepts, along with a more nuanced discussion of the studies limitations and implications. The lack of formal statistical analysis is also concerning. I have provided some more specific suggestions below, which I hope will be of use to the authors.

* When I initially read the manuscript, I assumed that when the authors referred to two groups, one group would be maintained on the usual diet, whereas the other would be exposed to a primarily plant based diet, however from what I understand, this is not the case and all birds were maintained on the same diet. Although the initial control diets do provide a clear point of comparison, the reliance on historical data to guide interpretation of body mass changes during the experimental diet could be considered a limitation of this study. Did you consider including a control group of birds maintained on the standard diet throughout? What was the purpose of randomization to the two groups? If simply for housing purposes, it may be useful to just explain this in the methods as the reference to two groups in the abstract may mislead some readers.

* In the abstract it is implied that the observation of some birds eating seagrass rhizomes is a relatively anomalous occurrence (Line 17), whereas in the main text, it is stated that plants may form a "sizeable" part of the diet, for some groups or individuals at least (Line 62). It would be useful to be more explicit about what "sizeable" means, e.g., are we talking about a meaningful and consistent element of total dietary intake, or just a frequent supplement? It is difficult to

reconcile the description of this species as "obligatory molluscivorous" if plants really do form a sizeable portion of the diet.

* Results and Discussion: The figures were useful to evaluate overall trends, however I found the results difficult to follow without quantification and statistical analysis. For example, terms such as "slightly heavier" and "somewhat lower" are open to interpretation, and it would be useful to be more objective in the results reporting. This is particularly relevant considering the large variation apparent in Figure 1.

* Based on visual analysis of the data presented in Figure 1, I am not entirely convinced by the conclusion that the studied red knots adapted well to the plant-based diet. Even though the mean data seem similar to the historical data, there is quite a noticeable drop in body mass throughout this time period, which quickly recovers as soon as they return to the control diet. Of course this is just my subjective evaluation of the average trends reported, and further to the above point, a more precise definition of the main comparisons of interest alongside more formal analysis would allow for a better-informed interpretation.

* Line 171 - 172: Please elaborate further on the birds initial response to the plant-based pellets. When you say they were "hesitant" do you mean that they approached and sampled them slowly, or that they avoided them entirely? Going by the information presented on Line 173 I believe it may have been the latter, if the group did not eat at all for the first 2-3 days and only began feeding once one individual initiated it. This would mean that the early weight loss and faecal changes may reflect short-term starvation rather than the nutritional properties of the pellets. When you state that the birds adapted within 2-3 days after they began eating on Line 175, does this reflect recovery by approximately day 5-6? It is possible that the patterns observed may reflect resumption of feeding rather than physiological adaptation to the new diet. I believe this distinction to be important, as the next paragraph proposes various gastrointestinal adaptations that may have occurred in response to the change to the plant-based pellets, but it is not entirely clear whether the observed changes were due to GI adaptation to the new diet, or whether they were driven by initial food avoidance due to unfamiliarity with the pellets.

* Line 81: What was the purpose of the initial blood sample and why were these results not reported?

* Line 106 - 110: Please provide further information on the length of each phase. This information is provided in the abstract, but not in the main text.

* Line 156 - 157: What did you mean when you said that weight increase following return to the control diet suggested deposition of (wet) protein rather than fat?

* Line 187: I was unclear on the point being made here, nor how the conclusion that body mass was abnormally high for the season and stage was reached. According to Figure 1 there did indeed seem to be a body mass increase immediately following the return to the control diet, but I'm not sure how this was determined to be "abnormally high" considering it was still well within the variation range of the historical data. Returning to the above point, more precise definition of the research questions and comparisons of interest, along with formal analysis, would be very useful to better understand these results.

* Figure 2: Is this correct? If I have understood well, the figure seems to indicate an abrupt increase in body mass and reduction in faecal green color on initiation of the plant-based diet, whereas the rest of the paper implies the opposite.

* The manuscript ended rather abruptly, and it may be useful to include a concluding section that comments on study limitations, implications and future directions. Considering limitations I do believe that the small group sizes and reliance on historical data should be commented on, as these factors may impact interpretation. The findings are also based only on body mass and faecal characteristics and it may be interesting to comment on additional parameters that could be evaluated in future work. For example, I wonder if these birds would manage to migrate on a largely plant-based diet? This study also aligns with ongoing human research related to animal

versus plant protein sources, where the emerging consensus appears to be that provided nutrient intake is adequate, the food source is less important.

Reviewer 2

Comments for the author

This study provides valuable insight into the dietary flexibility of red knots, a shorebird typically specialized in consuming shellfish. By shifting captive individuals from a protein-rich diet to a plant-based diet, the authors show that although the birds initially lost weight, they rapidly adjusted and stabilized at a slightly lower body mass. Changes in faecal colour and volume further suggest quick physiological and digestive adaptation. These findings demonstrate that red knots are capable of switching to plant material when environmental conditions require it, highlighting a potentially important mechanism for coping with habitat or prey changes.

The study is thoughtfully designed, and the manuscript is clearly written. I offer several suggestions that may help strengthen the clarity and impact of the work:

1. Please provide details on environmental conditions during the experiment (temperature regime, photoperiod, and salinity if relevant). These factors can influence metabolism and digestion.
2. The methods do not describe how body mass was measured (e.g., frequency, equipment used, handling procedures). This information is essential for assessing accuracy and repeatability.
3. Results and discussion are currently combined, which makes it difficult to distinguish observations from interpretation. Separating these sections would make findings clearer and improve readability.
4. The discussion includes several strong points but would benefit from clearer structure and smoother narrative flow. Consider grouping ideas into thematic paragraphs.
5. The study focuses on body mass and faecal characteristics, important but basic metrics. The brief mention of future directions could be expanded into a dedicated section outlining limitations and potential next steps (e.g., gut morphology, microbiome analyses, digestive enzyme activity).
6. Please ensure figure legend and axes are appropriate. Figure 1 appears to compare the experimental results with data from 2014-2021, yet the x-axis only displays dates from November 2023 to April 2024. Please revise the axis or clarify in the legend to avoid confusion.

Reviewer's Responses to Questions

Experimental quality

Does each figure have the proper controls?

If 'No', please indicate reasons in Comments for Author box below.

Reviewer #1:

- Yes

Reviewer #2:

- Yes

Were the data analyzed using appropriate statistical tests?

If 'No', please indicate reasons in Comments for Author box below.

Reviewer #1:

- No

Reviewer #2:

- Yes

Reproducibility

Were experiments performed using adequate number of biological replicates?
If 'No', please indicate reasons in Comments for Author box below.

Reviewer #1:

- Yes

Reviewer #2:

- Yes

Does the methods section provide sufficient detail to permit reproducibility?
If 'No', please indicate reasons in Comments for Author box below.

Reviewer #1:

- Yes

Reviewer #2:

- Yes

Completeness

Are the manuscript's conclusions supported by the data?
If 'No', please indicate reasons in Comments for Author box below.

Reviewer #1:

- No

Reviewer #2:

- Yes

Scholarship

Do the authors cite and discuss the merits of data that would argue for and against their conclusion?
If 'No', please indicate reasons in Comments for Author box below.

Reviewer #1:

- Yes

Reviewer #2:

- No

Does the manuscript title & abstract accurately reflect the contents of the manuscript, without hyperbole?
If 'No', please indicate reasons in Comments for Author box below.

Reviewer #1:

- Yes

Reviewer #2:

- Yes

First revision

Author response to reviewers' comments

Comments from the Reviewers:

Reviewer 1: The authors exposed a group of typically molluscivorous red knots (*Calidris canutus islandica*) to a primarily plant based diet. They reported that the studied birds rapidly adapted to the new diet, demonstrating dietary flexibility in this species.

Overall I found this to be a very interesting study and believe it is an important topic to explore, considering the important role that dietary flexibility may play in this rapidly changing world. The manuscript is generally well-written and parts are compelling. There were, however, some aspects that I found difficult to understand and in my opinion, the manuscript would be strengthened by clarification of some core concepts, along with a more nuanced discussion of the studies limitations and implications. The lack of formal statistical analysis is also concerning. I have provided some more specific suggestions below, which I hope will be of use to the authors.

RESPONSE

Fully realizing the limitations of our newly established findings, which is of course a feature of all studies, we are encouraged by the excitement of reviewer 1. Thank you.

Comm1* When I initially read the manuscript, I assumed that when the authors referred to two groups, one group would be maintained on the usual diet, whereas the other would be exposed to a primarily plant based diet, however from what I understand, this is not the case and all birds were maintained on the same diet. Although the initial control diets do provide a clear point of comparison, the reliance on historical data to guide interpretation of body mass changes during the experimental diet could be considered a limitation of this study. Did you consider including a control group of birds maintained on the standard diet throughout? What was the purpose of randomization to the two groups? If simply for housing purposes, it may be useful to just explain this in the methods as the reference to two groups in the abstract may mislead some readers.

RESPONSE

The lack of parallel control is indeed a limitation of this study and this will now be mentioned in line 219, thank you.

We considered to include a control group of birds that maintained on the standard diet. The study was intended as a pilot to study the effect of diet on the gut microbiome of red knots (through DNA analysis of faecal matter). The diet was not tested before and a maximum of 10 birds was decided, based on prior experience with experiments on red knots. Cohousing is expected to have a large effect on gut microbiome. Red knots have a circannual rhythm and having two groups switch diet on a different date seemed suboptimal due to unknown seasonal effects on the gut microbiome. We therefore opted to have both experimental groups transition at the same time. We will now include sample size information in line 171.

To serve as a real time control, we had the four trainer birds that remained on the control diet. We opted for historical data as a more reliable control for body mass as the four trainer birds had been in captivity for a long time and one was a different subspecies.

Randomization was indeed for housing purposes only, we did not want to be biased based on initial body mass, mould or plumage. It would indeed be good to mention this in the materials and methods, which we now did in line 170.

Comm2* In the abstract it is implied that the observation of some birds eating seagrass rhizomes is a relatively anomalous occurrence (Line 17), whereas in the main text, it is stated that plants may form a "sizeable" part of the diet, for some groups or individuals at least (Line 62). It would be useful to be more explicit about what "sizeable" means, e.g., are we talking about a meaningful and consistent element of total dietary intake, or just a frequent supplement? It is difficult to reconcile the description of this species as "obligatory molluscivorous" if plants really do form a sizeable portion of the diet.

RESPONSE

We completely agree and the distinction between a meaningful and consistent element or a just a frequent supplement is important to make, thank you.

Red knots do not prefer seagrass. In the field in Mauritania younger and shorter-billed individuals will resort to eat seagrass. The proportion of seagrass in the diet of red knots is unknown and expected to differ between individuals. We adapted line 68 to write this more nuanced and based on observations.

Comm3* Results and Discussion: The figures were useful to evaluate overall trends, however I found the results difficult to follow without quantification and statistical analysis. For example, terms such as "slightly heavier" and "somewhat lower" are open to interpretation, and it would be useful to be more objective in the results reporting. This is particularly relevant considering the large variation apparent in Figure 1.

RESPONSE

This is indeed open to interpretation, we therefore adjusted the figure and included statistics. This leads to considerable changes in the text in lines 95-103 and in the materials and methods section lines 223-232.

Comm4* Based on visual analysis of the data presented in Figure 1, I am not entirely convinced by the conclusion that the studied red knots adapted well to the plant-based diet. Even though the mean data seem similar to the historical data, there is quite a noticeable drop in body mass throughout this time period, which quickly recovers as soon as they return to the control diet. Of course this is just my subjective evaluation of the average trends reported, and further to the above point, a more precise definition of the main comparisons of interest alongside more formal analysis would allow for a better-informed interpretation.

RESPONSE

The visual analysis in Figure 1 indeed lacks a more precise definition of the main comparisons of interest. We included a more formal analysis that makes a prediction, based on the historical data, rather than comparing with the historical data. As mentioned in Comm3 this leads to considerable changes in the text in lines 95-103 and in the materials and methods section lines 223-232. In addition, we added this information in Figure 1 to make this visually more clear.

Comm5* Line 171 - 172: Please elaborate further on the birds initial response to the plant-based pellets. When you say they were "hesitant" do you mean that they approached and sampled them slowly, or that they avoided them entirely? Going by the information presented on Line 173 I believe it may have been the latter, if the group did not eat at all for the first 2-3 days and only began feeding once one individual initiated it. This would mean that the early weight loss and faecal changes may reflect short-term starvation rather than the nutritional properties of the pellets. When you state that the birds adapted within 2-3 days after they began eating on Line 175, does this reflect recovery by approximately day 5-6? It is possible that the patterns observed may reflect resumption of feeding rather than physiological adaptation to the new diet. I believe this distinction to be important, as the next paragraph proposes various gastrointestinal adaptations that may have occurred in response to the change to the plant-based pellets, but it is not entirely clear whether the observed changes were due to GI adaptation to the new diet, or whether they were driven by initial food avoidance due to unfamiliarity with the pellets.

RESPONSE

We agree that the initial drop in body mass is likely due to starvation only and it is very important that this is clear, thank you. We therefore mention it early on in the discussion, lines 121-123.

Comm6* Line 81: What was the purpose of the initial blood sample and why were these results not reported?

RESPONSE

The purpose of the initial blood sample is a routine practice to later determine the sex of the birds and we added this in the text in line 151, thank you. We did not use the sex of the birds in our analysis in this study because we do not expect an effect of sex on adapting to a plant diet.

Comm7* Line 106 - 110: Please provide further information on the length of each phase. This information is provided in the abstract, but not in the main text.

RESPONSE

The length of each phase is very important to mention indeed, thank you. We included this in the materials and methods in lines 180-183.

Comm8* Line 156 - 157: What did you mean when you said that weight increase following return to the control diet suggested deposition of (wet) protein rather than fat?

RESPONSE

The deposition of (wet) protein rather than fat was a speculation on the rapid body mass increase as a discussion point. However, this was meant as a brief suggestion and should not be causing confusion. We now removed this sentence from the text, thank you.

Comm9* Line 187: I was unclear on the point being made here, nor how the conclusion that body mass was abnormally high for the season and stage was reached. According to Figure 1 there did indeed seem to be a body mass increase immediately following the return to the control diet, but I'm not sure how this was determined to be "abnormally high" considering it was still well within the variation range of the historical data. Returning to the above point, more precise definition of the research questions and comparisons of interest, along with formal analysis, would be very useful to better understand these results.

RESPONSE

We completely agree that a more formal analysis is needed, thank you. In line with the previous comments, we added a more formal analysis and changes to Figure 1 and hope that our results and discussion are now more clear.

Comm10* Figure 2: Is this correct? If I have understood well, the figure seems to indicate an abrupt increase in body mass and reduction in faecal green color on initiation of the plant-based diet, whereas the rest of the paper implies the opposite.

RESPONSE

Figure 2 is indeed correct. The figure shows the first 10 days after the diet switch. Panel A shows the percentual change in body mass compared to the previous sampling point. During the first three days, the individuals lose weight (3 to 12 percent of their body mass) before increasing again.

Panel B shows indeed the relative amount of green colour observed in the faeces and one day after receiving the plant diet, the proportion of green colour is highest, thereafter it decreases.

We find it very important that this is clear to the reader and therefore changed the legend and the size of the x-axis, thank you.

Comm11* The manuscript ended rather abruptly, and it may be useful to include a concluding section that comments on study limitations, implications and future directions. Considering limitations I do believe that the small group sizes and reliance on historical data should be commented on, as these factors may impact interpretation. The findings are also based only on body mass and faecal characteristics and it may be interesting to comment on additional parameters that could be evaluated in future work. For example, I wonder if these birds would manage to migrate on a largely plant-based diet? This study also aligns with ongoing human research related to animal versus plant protein sources, where the emerging consensus appears to be that provided nutrient intake is adequate, the food source is less important.

RESPONSE

We agree that our findings are based on a limited amount of parameters and more research is needed to understand the possible mechanism that allow red knots to shift to a plant diet. We included the limitations of our study in the materials and methods in lines 219-221 and we included a statement about our future research in line 143, thank you.

Reviewer 2: This study provides valuable insight into the dietary flexibility of red knots, a shorebird typically specialized in consuming shellfish. By shifting captive individuals from a protein-rich diet to a plant-based diet, the authors show that although the birds initially lost weight, they rapidly adjusted and stabilized at a slightly lower body mass. Changes in faecal colour and volume further suggest quick physiological and digestive adaptation. These findings demonstrate that red knots are capable of switching to plant material when environmental conditions require it, highlighting a potentially important mechanism for coping with habitat or prey changes.

RESPONSE

We thank reviewer 2 for the kind words and constructive feedback on our article.

The study is thoughtfully designed, and the manuscript is clearly written. I offer several suggestions that may help strengthen the clarity and impact of the work:

Comm12 1. Please provide details on environmental conditions during the experiment (temperature regime, photoperiod, and salinity if relevant). These factors can influence metabolism and digestion.

RESPONSE

We agree that environmental conditions can influence metabolism and digestion, thank you. We have no exact details on temperatures, photoperiod and salinity. The aviaries used in this study are outside aviaries and the birds are exposed to outside air (temperature) and photoperiod. The salt water flow on the floor of the aviaries is sourced from the nearby Wadden Sea. We mention this in the material and method section in line 159-165.

Because we compare body mass to that of previous years during the same period of time, under similar circumstances (the same location, aviaries and diet), we are confident that we capture that variation in our study.

Comm13 2. The methods do not describe how body mass was measured (e.g., frequency, equipment used, handling procedures). This information is essential for assessing accuracy and repeatability.

RESPONSE

We agree that describing how body mass was measured is essential for assessing accuracy and repeatability. We now included this information in the materials and methods in line 191 -192, thank you.

Comm14 3. Results and discussion are currently combined, which makes it difficult to distinguish observations from interpretation. Separating these sections would make findings clearer and improve readability.

RESPONSE

We agree that it is important to distinguish observations from interpretations. We have now separated these sections, thank you.

Comm15 4. The discussion includes several strong points but would benefit from clearer structure and smoother narrative flow. Consider grouping ideas into thematic paragraphs.

RESPONSE

Thank you very much for the suggestion of grouping the ideas of the discussion into thematic paragraphs. We now have structured the ideas in the discussion section for a better flow, thank you.

Comm16 5. The study focuses on body mass and faecal characteristics, important but basic metrics. The brief mention of future directions could be expanded into a dedicated section outlining limitations and potential next steps (e.g., gut morphology, microbiome analyses, digestive enzyme activity).

RESPONSE

We agree that our findings are based on a limited amount of parameters and more research is needed to understand the possible mechanism that allow red knots to shift to a plant diet. We included the limitations of our study in the materials and methods in lines 219-221 and we included a statement about our future research in line 143, thank you.

Comm17 6. Please ensure figure legend and axes are appropriate. Figure 1 appears to compare the experimental results with data from 2014-2021, yet the x-axis only displays dates from November 2023 to April 2024. Please revise the axis or clarify in the legend to avoid confusion.

RESPONSE

A correct legend and clarification is indeed very important to understand the Figure. We have changed both the legend and the x-axis and are confident that the Figure has improved, thank you.

Second decision letter

MS ID#: bio.062365R1

MS TITLE: Molluscivorous red knots rapidly adjust to a plant diet

AUTHORS: Marie De Wilde; Berber Maarsingh; Luc de Monte; Anne Dekinga; Hidde Kressin; Allert I. Bijleveld; Theunis Piersma; Aldert L Zomer

This morning, I have read through your responses to the reviewer comments and the associated changes to your manuscript, and I am happy to tell you that your manuscript has been accepted for publication in Biology Open, pending our standard publication integrity checks. It was accepted on 2nd March 2026.